# Social Support and Commitment to Life and Living: Bidirectional Associations in Late Life over Time

**DOI:** 10.3390/healthcare11131965

**Published:** 2023-07-07

**Authors:** Sara Carmel, Norm O’Rourke, Hava Tovel, Victoria H. Raveis, Naama Antler, Ella Cohn-Schwartz

**Affiliations:** 1Department of Epidemiology, Biostatistics and Community Health Sciences, Faculty of Health Sciences, Ben-Gurion University of the Negev, Be’er Sheva 8410501, Israel; 2Center for Multidisciplinary Research in Aging, Faculty of Health Sciences, Ben-Gurion University of the Negev, Be’er Sheva 8410501, Israel; 3Department of Psychology, Faculty of Humanities and Social Sciences, Ben-Gurion University of the Negev, Be’er Sheva 8410501, Israel; 4Psychosocial Research Unit on Health, Aging and the Community, New York University College of Dentistry, New York, NY 10010-2314, USA

**Keywords:** commitment to living, fear of death, fear of dying, late life, will to live

## Abstract

Objectives: This study aims to enhance the understanding of longitudinal associations between two important facets of well-being in late life: social support and commitment to life and living (CTL). Methods: Structured home interviews were conducted with 824 Israelis ≥75 years of age, with three annual data collection timepoints. We hypothesized and tested a cross-lagged, longitudinal structural equation model (SEM) in which CTL and social support were assumed to predict each other over time, covarying for previously reported CTL and social support. Results: Social support has a positive, contemporaneous effect, predicting commitment to living at T1 and T3, while CTL predicts social support the following year (i.e., T1–T2 & T2–T3). Satisfaction with relationships significantly contributes to measurement of both latent constructs at each point of data collection. Discussion: Commitment to life and living and social support are intertwined phenomena. Whereas social support has a concomitant effect on CTL, the effect of CTL on social support emerges over time. This suggests that greater social support fosters greater CTL, leading older adults to nurture social networks and relationships; the effect of which is greater social support in the future. The implications of these results warrant further research over longer periods and across cultures.

## 1. Introduction

Understanding of well-being in late life is of particular importance, as this stage is associated with various age-related challenges and losses (e.g., decline in health and functioning, death of family and friends). Physical, social, and emotional challenges further exacerbate the decline in psychological reserves and resources including self-esteem, self-efficacy, subjective well-being, and even interest in life [1,2,3]. However, late life can also be associated with improvements in well-being and better emotional aging [4].

Positive psychology focuses on well-being and various intrinsic and external factors that foster physical and cognitive functioning, general well-being, and longevity [5]. Research has strived to promote empirically-based practical applications including psychosocial interventions and social policy [6,7]. Longitudinal research indicates a positive perception of aging fosters self-efficacy and physical functioning [8], and that will to live predicts depressive symptoms, not vice versa [9], and underscores both the theoretical importance and practical implications of this research.

Well-being in late life is composed of various factors including happiness, life satisfaction, perceived health, and will to live [10,11]. Associations across constructs are moderately but not strongly related, in part because each is comprised of multiple components, some of which are unique and others that are shared across constructs [12,13,14]. Understanding the effects of individual differences requires a multivariate study of predictors and interrelationships across facets of well-being over time. Consequently, in addition to the study of the effects of various resources on well-being, a deeper investigation of the interrelationships and influences among the various indicators of general well-being appears to be of importance. For instance, will to live appears to moderate the death-related decline in life satisfaction, indicating that psychosocial interventions intended to strengthen will to live appear effective up until the end of life [15].

For this study, we set out to determine the direction and strength of both contemporaneous and longitudinal associations between core facets of personal resources and well-being in late life; specifically, we studied social support and commitment to life and living (CTL). 

### 1.1. Will to Live

Like all mammals, humans are born with a strong instinct to survive (i.e., natural phenomenon). Will to live (WTL), however, is a biopsychosocial phenomenon comprised of both instinctual and psychological components; that is, WTL is as an existential need, drive, or purpose [10,16,17,18]. Societies and organized religion have a shared desire to perpetuate their existence by reinforcing the drive to continue living by fostering values such as sanctity of life and threats of eternal damnation for those who commit suicide [19].

The WTL is generally high for most people [20]. However, research also indicates variability in WTL in late life [10,16,21]. WTL appears to decline with age [10], and among those with severe illness [16]. WTL is also lower for women and those facing difficult health and life circumstances, such as those caring for a terminally ill relative [10,16].

WTL is as an important facet of well-being in late life, significantly associated with established indices of subjective well-being such as life satisfaction, self-esteem, successful aging, and fewer symptoms of depression [9,10,16,22]. Though distinct from other facets of well-being, WTL entails a desire to continue living even in the last chapter of life, a period characterized by new challenges, when adaptive resources are in decline [2,10]. In addition to serving as a facet of well-being, WTL has been reported to have a prognostic value. A study with Israeli older adults showed that WTL predicts survival after 7.5 years, especially among women, after controlling for both age and health [23]. Results of a 10-year Finnish study support this finding [24]. WTL is deeply rooted in the biopsychosocial existence of older adults who have more years behind them than ahead (i.e., approaching end of life).

### 1.2. Fear of Death and Dying

Death is frightening because it is the end of all that people live for and understand. Societies differ widely in ways they address such issues of purpose and meaning. Homans [25] argued that Christianity fosters fear of death (e.g., judgement, purgatory) but offers the promise of life after death for believers. Judaism focuses more on living than on life after death [18]; however, religious Jews tend to fear death more than secular Jews [26]. Denial of death along with religious customs serve as emotion-focused coping mechanisms that enable people to live blissfully as if death does not exist and will never come. 

Related to fear of death is fear of the dying process [27]. Due to scientific and technological advances, life can be prolonged even for those with severe and life-threatening diseases, often with pain, breathing difficulties, and other debilitating symptoms. Existing research suggests that suffering in the final stages of a terminal disease frightens people more than death itself [28]. WTL, fear of death, and fear of dying are independent and significant predictors of preferences for various life-sustaining treatments (e.g., artificial feeding, artificial ventilation, cardiopulmonary resuscitation) for those with severe illnesses [28]. Given the existential nature of each, we hypothesized that WTL and absence or limited fear of death and fear of dying constitute aspects of a higher-order construct we define as commitment to life and living (CTL). 

### 1.3. Social Support

To survive, grow, and develop, an additional basic need is support from others for much of human life, especially during periods of dependency (e.g., infancy) and often in late life [1]. The convoy model of social relations broadly defines social support as a multidimensional construct that includes aspects of practical and emotional support [29]. The perceived availability of social support pertains to the general availability of support as well as global satisfaction with the support received [30]. This construct captures perceived resources more accurately than more objective measures, which focus on exchange of support in specific contexts [31]. Consequently, subjective or perceived support is more strongly associated with mental health and well-being than instrumental support [32,33]. For this study, we define and operationalize social support as instrumental and emotional support and satisfaction with relationships with family and friends. 

The convoy model also emphasizes the importance of social support for physical and cognitive well-being in late life. Findings indicate direct and indirect effects of social support on emotional, physical, and cognitive health [34,35,36,37,38]. Social support is also associated with quality of life [39,40] and established indicators of subjective well-being [41], successful aging [42], and mental health [43]. To date, however, research linking social support and commitment to life and living is absent.

Due to the significance of social support on human survival, people tend to live with families and within various social groups and communities, generally based on common values and beliefs. Societies incorporate such groups into larger sociocultural organizations, which also strive to foster their ongoing physical and cultural existence. This leads societies to invest considerable resources in retaining and protecting their members, values, and norms of behavior [16]. One of the means to achieve these goals is the socialization process of new generations, during which societies invest in nurturing various facets of individuals’ lives through basic societal values and norms of behavior, including the biopsychosocial aspects of self-existence such as WTL and fear of death, as well as through societal cohesion in the form of values referring to social solidarity and mutual support. Societal influences on peoples’ will to continue living [44] or to sacrifice their lives for others are well known (e.g., wartime). Less is known about associations between social support and commitment to life and living despite its importance in late life. It is possible that people with better social support will also be more committed to living because of the positive well-being implications of social resources [45] and because they will have someone to “live for”. 

Considering the theoretical significance of both CTL and social support especially in late life, we examined associations between social support and CTL measured at three time points over two years. Since social support is an important predictor of well-being, we hypothesize that it also predicts CTL, a related yet different facet of well-being that has important implications to life in late life. We also predict that older adults with a stronger CTL will have more motivation to invest in their social ties. Thus, we hypothesized that the latent constructs of CTL (i.e., WTL, fear of death and dying) and social support (i.e., emotional support, instrumental support, and satisfaction with social relationships) are associated contemporaneously and over time. That is, we hypothesized that older adults reporting higher social support will report greater CTL concomitantly and in the future. Those reporting greater instrumental support, emotional support, and satisfaction with relationships will report higher CTL concomitantly and over time. 

## 2. Method

### 2.1. Participants and Procedures

Prospective participants were randomly identified from the population registry, stratified by gender and city of residence (32% in Haifa, 32.1% in Tel Aviv, and 35.3% in Be’er Sheva). We identified those ≥75 years of age living in the community, independent in activities of daily living (ADL), and without significant cognitive loss (i.e., a Mini-Mental State Examination score > 19/30; [46]). Men were oversampled at recruitment to account for greater mortality.

A total of 1216 older adults were interviewed at baseline, 1019 1 year later (T2; 83.8% of the original sample), and 892 at Time 3 (73.4%). Of those lost to attrition, 113 declined further participation, 131 became unable to participate (i.e., physical or cognitive disability), 60 died, and 20 could not be located despite repeated attempts. After excluding an additional 68 with missing socio-demographic data, 824 participants provided responses at each point of data collection (67.8% of the original sample). Those reporting a lower will to live at recruitment were more likely to be lost to attrition (e.g., deceased).

Structured interviews were conducted at three annual points of data collection over 2 years. All participants were interviewed in their homes by trained research assistants, sensitive to the fatigue and concentration levels of older adults. Interviews were conducted in either one or two sessions, lasting 1.5 to 3 h. Written consent was obtained from participants. 

### 2.2. Measures

The Will to Live Scale (WTL) [10] is composed of five items with Likert responses ranging from *no WTL* (0) to *very strong WTL* (5). Internal consistency of responses by older adults is high (0.81 < α < 0.90) [10,17]). For this study, internal consistency is ideal at each point of data collection (0.83 < α < 0.90). 

The Fear of Death Scale [28] is composed of six items measuring fear of the unknown, judgement, and eternal life (e.g., “I am very afraid of death”; “The fact that death means the end of everything that I know frightens me greatly”). In contrast, the 6-item Fear of Dying Scale measures fears regarding sever conditions at the end of life (e.g., “I am afraid of the suffering related to dying”; “I am afraid of losing respect at the end of my life”). Responses to both are measured on a Likert scale ranging from *completely disagree* (1) to *completely agree* (5). Adequate internal consistency for fear of death (α = 0.71) and fear of dying (α = 0.76) was initially reported with a representative sample of older Israelis ≥70 years of age (*N* = 1136) [28]. For the current study, internal consistency is moderate to high at each point of data collection for fear of death (0.77 < α < 0.78) and fear of dying (0.84 < α < 0.89). See Table 1. 

The Berlin Social Support Scales (BSSS) [47] measure both emotional support (four items; e.g., “Every time I feel bad different people show me that they like me”) and instrumental support (four items; e.g., “Some people offer me help when I need it”). Responses were reported on a Likert scale ranging from *not true at all* (1) to *absolutely true* (5). Internal consistency was high at each point of data collection for both emotional (0.82 < α < 0.88) and instrumental support (0.86 < α < 0.91). See Table 1. 

Satisfaction with social relationships was measured by two questions: “How satisfied are you with your relationships with friends, acquaintances, or neighbors?” and “How satisfied are you with your family relationships?” Responses to both were provided on a Likert scale ranging from *not at all satisfied* (1) to *very satisfied* (5) and the two items were averaged. 

A socio-demographic questionnaire was created to collect descriptive information (e.g., date of birth, gender, religiosity, household composition). Education was measured as primary, secondary, post-secondary, and academic. Self-rated health was assessed by a single question using a Likert scale from *excellent* (0) to *very bad* (5). 

### 2.3. Statistical Analyses

We hypothesized a longitudinal, structural equation model (SEM) in which social support contemporaneously predicts CTL at each point of measurement; and social support and CTL predict each other in the future (i.e., fully cross-lagged SEM). We covaried for prior measurement of both latent constructs so that additional variance explained across constructs is unique. Error was assumed to be correlated across the same item pairs over time. Various socio-demographic variables associated with social support and CTL were first examined and included as covariates if significantly associated with either latent construct. Descriptive statistics were computed with SPSS v28; the SEM was computed with AMOS v26.

Consistent with convention, we report three goodness-of-fit-indices: An incremental, an absolute, and a parsimonious fit index. The Comparative Fit Index (CFI) is an incremental index representing the extent to which a hypothesized model is a better fit to data than the null model. The Standardized Root Mean Square Residual (SRMR) is an absolute index that represents the standardized difference between observed and predicted correlations within a hypothesized model. Finally, the Root Mean Square Error of Approximation (RMSEA) is a parsimony index that represents the extent to which a hypothesized model fits data relative to the population. Coefficient values greater than 0.94 for the CFI, and less than 0.055 for the SRMR and RMSEA indicate good model fit [48].

## 3. Results

At recruitment, participants were 80.1 years of age on average (SD = 3.93, range 75–96 years) and most were men (n = 449, 54.5%). The plurality was born in Central or Eastern Europe (e.g., Poland, n = 286, 34.7%) followed by Israel (n = 146, 17.7%), Asia/Africa (e.g., Morocco, n = 135, 16.4%), the former USSR (n = 136, 16.5%), Western Europe (n = 72, 8.7%) and North America/Australia (n = 46, 5.6%). Most reported Hebrew as their primary language (n = 712, 86.4%) though a percentage continue to speak mostly Russian (n = 112, 13.6%). A high percent of older immigrants from the former Soviet Union remain more proficient in Russian than Hebrew.

Most participants had completed college (n = 145, 17.6%) or university (n = 295, 35.8%). Most described themselves as secular (n = 532, 64.6%), and a minority self-described as traditional (n = 245, 30.8%), orthodox (n = 37, 4.5%), or ultra-orthodox (n = 1, 0.1%). Most were widowed (n = 652, 79.1%); only a minority were currently married/partnered (n = 113, 13.8%), divorced (n = 34, 4.1%), or never married (n = 12, 1.5%).

### Structural Equation Modeling 

Consistent with previous findings, education was inversely associated with social support [49], and perceived health predicted CTL; that is, older adults reporting better health are more committed to life and living. Neither sex, religiosity, living arrangement, nor age emerged as significantly associated with either latent construct at baseline. The latter may be due to the restricted age range of participants at recruitment (i.e., 75–96 years of age). Perceived health and education were included as covariates at baseline.

The SEM that emerged differed from the fully cross-lagged model we hypothesized. After correcting for the correlated error between six item pairs (i.e., different items), the goodness of fit for this model was ideal, χ^2^ (*df* = 137) = 324.42, *p* < 0.01. More specifically, CFI = 0.97, SRMR = 0.053, and RMSEA = 0.041 (0.035 < RMSEA CL_90_ < 0.047). With 824 participants and 137 degrees of freedom, this model has sufficient statistical power (*d* = 0.99) to identify medium to small effect sizes (where α = 0.05) [48].

Each social support variable (i.e., emotional support, instrumental support, satisfaction with social relationships) contributed significantly to measurements at each point of data collection (i.e., CR > |1.96|). The same occurred for each of the three CTL variables (WTL, fear of death/dying). However, satisfaction with relationships significantly cross-loaded between latent constructs at each point of measurement; thus, it contributed to measurements of both social support and CTL (see Figure 1).

Social support at baseline predicted social support measured one and two years later. By contrast, CTL was predicted only by CTL measured the year prior (i.e., Time 1 did not predict Time 3). As hypothesized, social support concomitantly predicted CTL at baseline and T3, and CTL predicted social support when next reported (T1–T2 and T2–T3). However, the reverse longitudinal effect did not emerge. That is, CTL predicts future social support but not vice versa. The effect of social support on CTL appears contemporaneous only. These cross-lagged effects emerged while covarying for latent constructs over time. 

## 4. Discussion

Commitment to life and living is a predictor of well-being in late life [10,16]. As social support is integral to biopsychosocial development and general well-being [50], we hypothesized bidirectional associations of CTL and social support in late life over time; we set out to assess the direction of these associations. Using SEM, we identified contemporaneous and longitudinal associations between social support and CTL among older Israelis living in the community. We found that CTL was related to higher social support over time, while social support was not related to future CTL. 

Those more committed to life and living reported greater social support the following year at both time points. It could be that those more committed to life are more likely to actively invest in maintaining and cultivating familial and social relationships. The benefit of this investment in social relationships is greater social support the following year. These findings are in accord with socioemotional selectivity theory [51], which contends that as time narrows in late life, older adults adjust their social priorities to focus more on emotionally close ties. Our findings suggest that greater CTL, in addition to one’s sense of how long they have to live, foster social connectedness in late life. 

Our findings suggest that social support and CTL are interconnected in late life. That is, social support appears to have a positive, contemporaneous effect on CTL, indicating that those who report higher availability of emotional and instrumental support, and who are satisfied with their social relationships, are more strongly committed to life and living. This can be explained by the positive functions of emotional and instrumental support that make life easier and more meaningful, as they contribute to a sense of belonging, being cared for, and protection [52]. In accord with the convoy model [29], these findings demonstrate the importance of perceived social support to CTL and future social support. 

Furthermore, satisfaction with relationships (family and friends) contributes to measurements of both social support and CTL (i.e., cross-loads across latent constructs at each of the three points). That is, satisfaction with relationships is directly related to CTL. Satisfaction with relationships occurs when social networks meet one’s expectations. This appears especially important in relation to age-related losses in late life. Previous findings indicate that perceived quality of life and WTL depend on attainment of psychosocial and spiritual needs [16,40]. This reminds us that each person “… is more than a simple organism struggling to survive” [53] (p. 163). In addition to perceived emotional and instrumental support, people expect their social relationships to provide enjoyment, leisure, and most importantly, a sense of belonging that is crucial for identity, self-esteem, worth, and meaning in life [54].

Although previous research reports that social support from friends and family is an important resource that older adults generally value and seek to nurture and maintain [55], the contemporaneous effects of social support on CTL, and the reverse, longitudinal effect of CTL on social support, demonstrate more complex associations over time between these constructs in late life. Our model and findings further indicate that satisfaction with social relationships specifically assists older adults in maintaining their CTL. 

These findings are in accord with qualitative research on Israeli older adults who were asked, “What strengthens your will to live?” Emerging themes were social networks, religious faith, honor versus ageism, and work/volunteering, but not health or functional ability as would be expected [56]. Collectivism and social solidarity were founding ideals of the Jewish State; a perceived decline in societal attachment and solidarity undermines WTL for this cohort of older Israelis [57]. The question arises as to whether these findings are characteristic only of older Israelis, or whether our model is broadly applicable to late life. 

### Limitations and Future Research

Although we initially identified a large, random sample of older Israelis, participants lost to attrition were in worse health and of lower social status. Caution should be taken when generalizing the results. Future research is needed to replicate this model with other samples and followed over longer time periods (e.g., if possible until end of life) in Israel and other societies. 

## 5. Conclusions

Results from this study highlight the complex, longitudinal associations between CTL and social support. Our findings indicate that the direction of association changes over time: Social support contemporaneously predicts CTL; and CTL predicts social support over time. An extended investigation of contemporaneous and longitudinal associations and effects among these and other aspects of general well-being is needed to further inform both theory and practice.

Over-and-above, objective or instrumental and emotional support and subjective appraisal of social relationships with family and friends significantly contributes to measurements of both social support and CTL at each point of measurement. It appears that the importance of subjective evaluation of social relationships extends to social support as well as to CTL. These findings underscore the interplay between various facets of personal resources and general well-being over time. Additionally, the results of this study underscore the importance of psychosocial interventions intended to maintain and promote well-being in late life. Such interventions should strive to foster commitment to life and living, which, in addition to a direct effect of fostering well-being, can also counteract the loss of social resources that commonly occurs in late life. 

## Figures and Tables

**Figure 1 healthcare-11-01965-f001:**
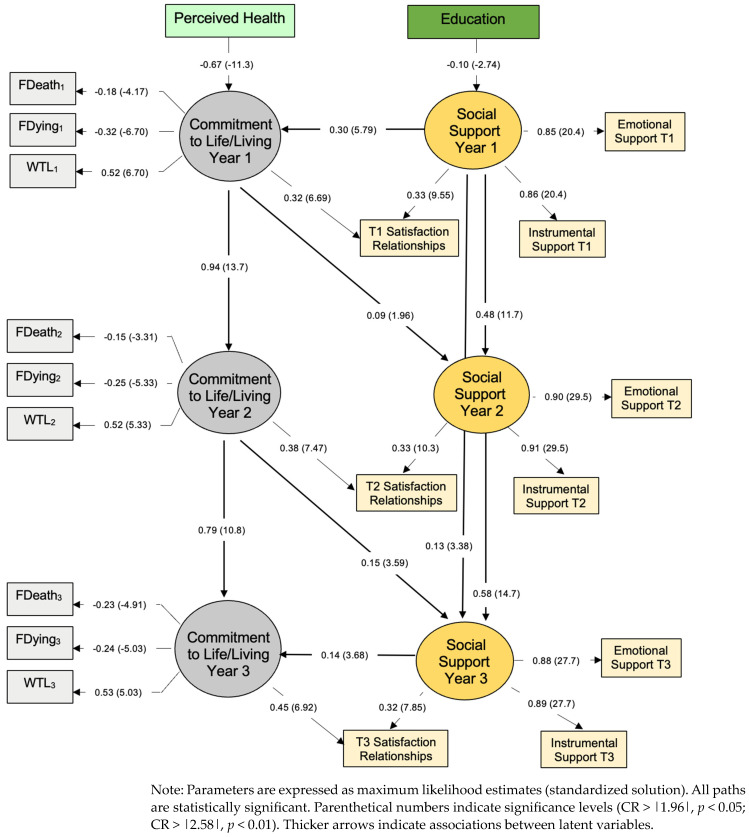
Social Support and Commitment to Life and Living over 2 years, Israelis ≥75 Years of Age.

**Table 1 healthcare-11-01965-t001:** Description of study variables and indices at T1, T2 and T3, Israelis, aged ≥75 (*N* = 824).

Variables		Mean	SD	Skewness	Kurtosis	α
Will to live [10]	T1	17.90	3.68	−0.89	2.02	0.83
	T2	17.77	4.05	−0.99	2.16	0.83
	T3	17.72	4.67	−0.94	1.34	0.90
Fear of death [28]	T1	9.14	4.44	1.99	4.25	0.77
	T2	9.12	4.40	2.01	4.57	0.77
	T3	8.93	4.38	2.18	5.14	0.78
Fear of dying [28]	T1	22.05	6.99	−0.61	−0.80	0.84
	T2	22.04	7.30	−0.64	−0.87	0.87
	T3	22.55	7.39	−0.70	−0.78	0.89
Social support—emotional [47]	T1	17.43	3.23	−1.27	0.97	0.82
	T2	17.62	3.22	−1.46	1.85	0.86
	T3	17.63	3.27	−1.69	2.61	0.88
Social support—instrumental [47]	T1	17.24	3.67	−1.45	1.63	0.86
	T2	17.52	3.57	−1.64	2.44	0.90
	T3	17.68	3.60	−1.81	2.97	0.91
Satisfaction with social support [47]	T1	8.65	1.43	−1.17	1.21	--
	T2	8.59	1.51	−1.20	1.44	--
	T3	8.61	1.56	−1.44	2.37	--

Note: Cronbach’s alpha (α) was not computed for Satisfaction with social support as this variable is composed of only two items.

## Data Availability

Anonymized data available from the corresponding author on request.

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
