# Peer review of "Social Support and Commitment to Life and Living: Bidirectional Associations in Late Life over Time"

_healthcare, 2023, doi:10.3390/healthcare11131965_

Round 1
Reviewer 1 Report
This is a reasonably solid paper in many respects, I like the SEM approach to cross lagged longitudinal data. It would however benefit from several changes, some of which likely impact the paper's findings..
1. The description of the sample belongs in the methods section, not results.
2. More information is needed regarding the validity of the measures of CTL and social support (especially the former, given that the scale was developed by the author. It does not necessarily follow that persons higher in CTL are less death anxious. Indeed, much work suggests that it is an awareness of one's death that gives meaning to life. The literature on death and dying, as well as the measurement of the fear of each, needs expanding. That which is cited is very limited. Why did the authors not rely on established measures of fear of death and dying (e.g., Templer, Collett & Lester)? There are also established indices of social support available (e.g. Krauss).
3. I think that the rationale for the paper needs strengthening-what is it that is unique that this paper is contributing to our knowledge about CTL and about social support in later life? The fact that work linking CTL and social support is absent is not a strong rationale.
4. There needs to be a rationale for selecting only persons who are age 75 and older-clearly this biases the sample. Are findings generalizable for both men and women?
5. While attrition is discussed, its impact on social support and CTL at each occasion is not. Likewise, while it is stated that there were attrition effects for health and social status (not clearly defined), no findings were presented. Importantly, attrition effect for social support and for CTL were not reported either-this bias might help explain the lack of predictability over time for social support and for CTL, to say nothing of the lack of cross-lagged bidirectional associations between these two constructs.
6. The figure could use some revisions-it for example is not clear to what the parenthesized values refer-it is stated that that they refer to significance levels for each parameter, but they are not presented in a traditional manner (e.g., p < .05). If they are indeed CR values, this should be clarified. I think greater clarity is needed in the figure in differentiating paths which are/are not statistically significant.
7. The paper is densely written and most likely too long-the writing needs tightening. I also think the discussion might be somewhat overwritten-what are the implications for future research and for practice? Perhaps a refocused discussion would be more meaningful. Also, the limitations section (e.g. attrition effects, validity of the key measures, sampling bias) needs to be expanded upon.
Author Response
Dear Reviewer,
Response to reviewer comments:
1. The description of the sample belongs in the methods section, not results.
Response: Disciplines differ somewhat in relative structure of manuscripts. In social science research, for instance, it is common to begin the results section by describing the sample, especially with longitudinal research (i.e., resulting sample after attrition). We choose to leave as is.
2. More information is needed regarding the validity of the measures of CTL and social support (especially the former, given that the scale was developed by the author. It does not necessarily follow that persons higher in CTL are less death anxious. Indeed, much work suggests that it is an awareness of one's death that gives meaning to life. The literature on death and dying, as well as the measurement of the fear of each, needs expanding. That which is cited is very limited. Why did the authors not rely on established measures of fear of death and dying (e.g., Templer, Collett & Lester)? There are also established indices of social support available (e.g. Krauss).
Response: The reviewer is correct; there are conflicting results regarding the awareness of death and dying and the well-being in later life. Existential research suggests as positive association whereas our findings indicate the contrary at each point of measurement. There are many well-validated measures of social support for use with older adults. Psychometric findings (e.g., Cronbach’s alpha) support our choices.
3. I think that the rationale for the paper needs strengthening-what is it that is unique that this paper is contributing to our knowledge about CTL and about social support in later life? The fact that work linking CTL and social support is absent is not a strong rationale.
Response: The reviewer is again correct; there is limited theory regarding the construct of commitment to life and living in relation to social support in late life. This study was conducted to address this void.
4. There needs to be a rationale for selecting only persons who are age 75 and older-clearly this biases the sample. Are findings generalizable for both men and women?
Response: Older adults are increasingly heterogenous as growing numbers of people live into their 90s and beyond. For instance, those 90+ might well have children who are also 65+; that is, they constitute different generations. Far from a limitation as this comment suggests, restricting recruitment to those 75+ is a definitive study strength – especially for longitudinal research.
5. While attrition is discussed, its impact on social support and CTL at each occasion is not. Likewise, while it is stated that there were attrition effects for health and social status (not clearly defined), no findings were presented. Importantly, attrition effect for social support and for CTL were not reported either-this bias might help explain the lack of predictability over time for social support and for CTL, to say nothing of the lack of cross-lagged bidirectional associations between these two constructs.
Response: The reviewer is correct; those reporting lower will to live were less likely to provide responses at successive points of measurement. That is, now stated in the text.
6. The figure could use some revisions-it for example is not clear to what the parenthesized values refer-it is stated that that they refer to significance levels for each parameter, but they are not presented in a traditional manner (e.g., p< .05). If they are indeed CR values, this should be clarified. I think greater clarity is needed in the figure in differentiating paths which are/are not statistically significant.
Response: As now stated in the note to Figure 1, all paths are statistically significant. Critical ratio (CR) values greater than 1.96 are significant at p < .05 whereas paths with CR values greater than 2.58 are significant at p < .01.
7. The paper is densely written and most likely too long-the writing needs tightening. I also think the discussion might be somewhat overwritten-what are the implications for future research and for practice? Perhaps a refocused discussion would be more meaningful. Also, the limitations section (e.g. attrition effects, validity of the key measures, sampling bias) needs to be expanded upon.
Response: It is unclear why ‘densely written’ text needs ‘tightening’. Is this not contradictory? Other reviewers describe the discussion and limitations sections as thorough and well written.
For more details, please see the revised manuscript.
Reviewer 2 Report
Review of Social support & commitment to life and living: Bidirectional associations in late life over time
This paper described findings from a longitudinal study of 824 Israelis aged 75+ interviewed at home three times over a 2 yr period about social support and commitment to life and living (CTL). CTL is a new higher-order construct built from Will to Live (an existential need, drive, purpose) and absence or limited fear of death or fear of dying. Social support is another key construct in this work. The two have not been simultaneously explored and linked before. It was hypothesized that social support would predict CTL at each point of measurement and social support and CTL would predict each other in future.
Strengths: large geographically diverse sample (about a third of which came from each of the three cities of Haifa, Tel Aviv and Beer Sheva); measurement of key variables with psychometrically sound scales (Berlin Social Support Scales which measure both emotional and instrumental support, Will to live scale, Fear of Death Scale and Fear of Dying Scale) plus two items measuring satisfaction with social relationships. Sophisticated technique (Structural equation modelling) was used to analyze the data. Article is well written, clear and easy to read and understand. Interesting findings (CTL was related to higher social support over time; and contrary to hypothesis, social support was not related to future CTL). Good discussion of findings and recognition of study limitations.
Weaknesses: none
Author Response
There are no comments here that require revisions to the text.
Reviewer 3 Report
Thank you very much for offering me the opportunity to review this very interesting and of high quality paper. The authors investigated the complex relationship between social support and commitment to life and living. A large sample size was recruited and appropriate analyses were conducted. Some very minor suggestions are listed below:
General comments:
· The introduction might benefit from shorter sentences which will enhance the readers’ understanding.
1.1 Commitment to Life and Living
· Further explanation is needed on how WTL is a biopsychosocial phenomenon for the individual and not only within the society and why it is “deeply rooted” in older adults in particular.
Author Response
Dear Reviewer,
Response to reviewer's comments:
1. General comments: The introduction might benefit from shorter sentences which will enhance the readers’ understanding.
Response: More periods are now used; sentences are now shorter.
2. Commitment to Life and Living: Further explanation is needed on how WTL is a biopsychosocial phenomenon for the individual and not only within the society and why it is “deeply rooted” in older adults in particular.
Response: Will to live is more ‘deeply rooted’ in older adults as they are approaching end of life. This is now directly stated in the text.
For more details, please see the revised manuscript.
Reviewer 4 Report
1. In the background section, 1.1 "commitment to life and living" should be changed to "will-to-live", and this section mainly introduces "will-to-live".
2. Please explain why you chose the elderly aged 75 or above as the target of the survey, and what is the significance.
3. Please explain how randomness was maintained in the sampling process.
4. Table 1 should be in the results section rather than the methods section.
5. The discussion section did not seem to explain the role of perceived health, education on the relationship between social support and CTL.
6. Some references are too old and should be updated.
English language needs further improvement.
Author Response
Dear Reviewer,
Response to reviewer's comments:
1. In the background section, 1.1 "commitment to life and living" should be changed to "will-to-live", and this section mainly introduces "will-to-live".
Response: We have made this change.
2. Please explain why you chose the elderly aged 75 or above as the target of the survey, and what is the significance.
Response: Most participants recruited for gerontology research are young-old (i.e., 65-74 years of age), so restricting participation to those 75+ is a study strength. This is especially true for longitudinal research with older adults.
3. Please explain how randomness was maintained in the sampling process.
Response: As we state in the text, prospective participants were randomly identified, not randomly recruited. We do not contend that the resulting sample is representative, especially due to attrition over time.
4. Table 1 should be in the results section rather than the methods section.
Response: As we reference psychometric information in the methods section reported in Table 1 (i.e., Cronbach’s alpha), the table should remain as is.
5. The discussion section did not seem to explain the role of perceived health, education on the relationship between social support and CTL.
Response: We describe the covariates (health, education) in the results. These findings are in accord with existing research are do not warrant further discussion.
6. Some references are too old and should be updated.
Response: We include both new and older references to thoroughly present existing research. Scales were developed and initially validated many years ago. This is a study strength (i.e., not new, not-yet-validated instruments).
For more details, please see the revised manuscript.